# Peta-bit-per-second optical communications system using a standard cladding diameter 15-mode fiber

Georg Rademacher [1]✉, Benjamin J. Puttnam[1], Ruben S. Luís[1], Tobias A. Eriksson[1,2], Nicolas K. Fontaine [3], Mikael Mazur[3], Haoshuo Chen[3], Roland Ryf[3], David T. Neilson[3], Pierre Sillard[4], Frank Achten[5], Yoshinari Awaji[1] & Hideaki Furukawa[1]

Data rates in optical fiber networks have increased exponentially over the past decades and core-networks are expected to operate in the peta-bit-per-second regime by 2030. As current single-mode fiber-based transmission systems are reaching their capacity limits, space-division multiplexing has been investigated as a means to increase the per-fiber capacity. Of all space-division multiplexing fibers proposed to date, multi-mode fibers have the highest spatial channel density, as signals traveling in orthogonal fiber modes share the same fiber-core. By combining a high mode-count multi-mode fiber with wideband wavelength-division multiplexing, we report a peta-bit-per-second class transmission demonstration in multi-mode fibers. This was enabled by combining three key technologies: a wideband optical comb-based transmitter to generate highly spectral efficient 64-quadrature-amplitude modulated signals between 1528 nm and 1610 nm wavelength, a broadband mode-multiplexer, based on multi-plane light conversion, and a 15-mode multi-mode fiber with optimized transmission characteristics for wideband operation.

[1] Photonic Network Laboratory, Photonic ICT Research Center, National Institute of Information and Communications Technology, Koganei, Tokyo, Japan. [2] Infinera, Stockholm, Sweden. [3] Nokia Bell Labs, New Providence, NJ, USA. [4] Prysmian Group, Billy Berclau, Haisnes Cedex, France. [5] Prysmian Group, Eindhoven, CA, The Netherlands. ✉email: georg.rademacher@nict.go.jp

Space-division multiplexing (SDM)[1] is a promising technology to overcome the capacity limitations of current single-mode fiber (SMF)-based optical communications systems[2] and to satisfy the exponentially increasing capacity demands by transmitting different data streams over spatially diverse transmission paths[3]. While submarine transmission systems have begun to adopt SDM techniques, such as shared amplification schemes[4], further integration including the transmission medium is increasingly important to optimize space usage and to increase the cost- and energy-efficiency[3].

Different fiber types have been proposed for SDM, including coupled-core[5] and weakly coupled[6] single-mode multi-core fibers (MCF), few-[7] and multi-mode fibers (MMF)[8,9], and their hybrid combinations few-mode multi-core fibers[10]. Until now, all transmission demonstrations that transmitted data rates of more than 1 peta-bit/s in a single fiber have relied on optical fibers with cladding diameters in excess of the current 125 μm standard[11–19]. Maintaining the current standard for cladding diameters is likely to be advantageous for economic reasons, as current cabling technologies may be applied. Moreover, increasing the cladding diameter may also alter technical fiber properties such as the mechanical reliability, coupling tolerances, and production yield[20,21]. Hence, an increased interest has recently emerged in investigating SDM solutions where the industry standard for the cladding diameter can be maintained. While promising transmission demonstrations have been made using a weakly coupled 4-core MCF[22,23], it is evident that the highest spatial channel density and thus the largest number of spatial channels in a standard cladding diameter fiber can be achieved in MMF[9] as parallel spatial channels share the cross-section of the fiber-core. So far, MMF-based transmission systems have been demonstrated with data rates of 280 Tb/s[24] and 402 Tb/s[25].

In this work, we report a transmission demonstration exceeding 1 Pb/s in a 15-mode fiber with 125 μm cladding diameter, extending our previous publication[26]. Figure 1 illustrates the system setup for the experimental demonstration. A single optical comb source generated 382 comb lines with a 25 GHz spacing, spanning the C- and L-bands. The comb lines were modulated to produce $15 \times 382 \times 24.5$ GBaud dual-polarization 64-quadrature amplitude modulated (DP-64-QAM) signals for a total of 11,460 wavelength-division multiplexed (WDM)-SDM data channels. The signals were then transmitted over the 15 spatial fiber modes, while spatial multiplexing was achieved through mode-selective mode-multiplexers. The 15 output signals from the mode de-multiplexer were received by a 15-channel SDM receiver. As the 30 transmitted signal tributaries (15 spatial modes × 2 polarizations) of each WDM channel were subject to mode mixing, coherent $30 \times 30$ multiple-input/multiple-output (MIMO) equalization was included in the offline digital signal processing (DSP). The wideband transmission channel properties and total data rate reported in this paper highlight the large potential of high mode-count MMF for future high-capacity SDM optical transmission systems.

## Results

**Multi-plane light conversion mode multiplexer**. The mode-selective spatial multiplexer and de-multiplexer were based on multi-plane light conversion[27,28]. The principle of operation is shown in Fig. 2a. A one-dimensional input spot array was produced by an array of 15 SMF and subsequently transformed by 15 reflections between phase masks and a dielectric mirror to match the fiber modes of the 15-mode MMF. While the transformation can be loss-less when assuming ideal phase masks, fabrication limitations such as pixelated phase masks and a discretization of phase values lead to a theoretical loss of each multiplexer of approximately 0.3 dB. In addition, an excess loss of 0.25 dB can be assumed for each reflection on a phase mask due to scattering of light to higher order modes that are not supported by the transmission system, blurring of neighboring pixel values, and limited reflection on the dielectric mirror, for a total loss of 3.5 dB[27]. Figure 2b shows the measured wavelength-dependent average, minimum and maximum insertion loss for all 15 ports of one mode-multiplexer. The lowest average insertion loss was 9.2 dB at 1555 nm wavelength. We assume that the discrepancy between expected and measured loss stems from optical mis-alignment within the mode-multiplexer. Figure 2b also shows the difference between the highest and lowest modal loss of 2.5 dB at 1610 nm wavelength and 3.2 dB at 1530 nm wavelength. While this metric can serve as an indication of the mode-dependent loss (MDL) behavior, it should not be confused with MDL calculated from the channel transfer matrix, as presented later in this manuscript. A photograph of one of the two used multiplexers is shown in Fig. 2c.

**15-mode graded-index multi-mode fiber**. The 15-mode MMF[29] was 23 km long and had a trench-assisted, graded-index profile with a core radius of 14.1 μm and a core-cladding refractive-index difference (Delta) of 1% as shown in Fig. 3a. This refractive index difference was similar to that of standard MMF with a 50 μm core diameter. Therefore, standard trench-assisted graded-index multi-mode preforms were used to realize the fiber. It was necessary to adjust the graded-index core exponents (alpha) and trenches to minimize the differential mode group delay (DMGD) at the target wavelength around 1550 nm. The preform diameter was set at 1.77 times larger than that of standard multi-mode preforms in order to ensure a core radius of 14.1 μm in the drawn fiber. This limited the propagation to 15 spatial modes at 1550 nm

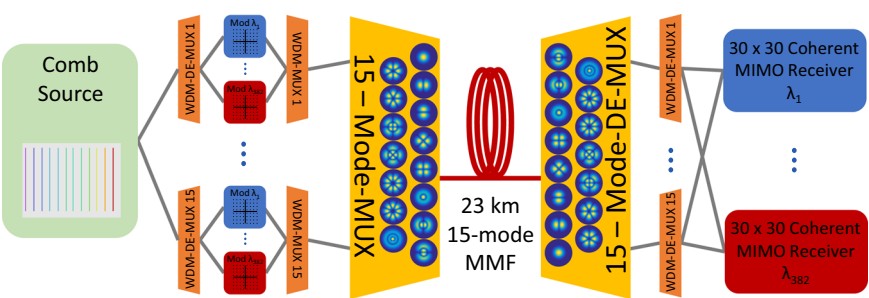

**Fig. 1 Schematic diagram of the optical transmission system.** A total of 382 laser lines were generated by an optical comb source to generate 15 × 382 independent dual-polarization 64-QAM data signals. Each set of 382 WDM signals was spatially multiplexed on a different fiber mode and jointly transmitted over a 15-mode fiber. After spatial de-multiplexing, each spatial super-channel, formed by the 15 spatial channels of the same wavelength, was received by a 30 × 30 coherent MIMO receiver.

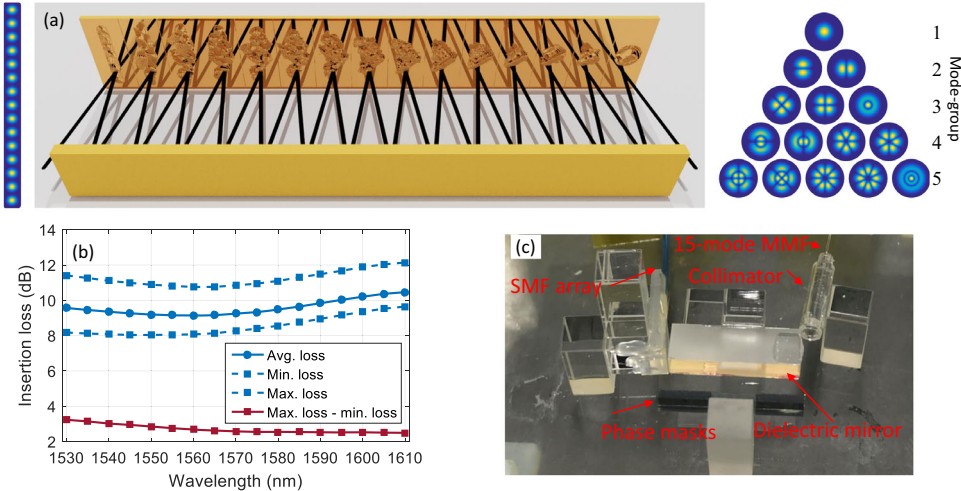

**Fig. 2 Mode-multiplexer based on multi-plane light conversion. a** Principle of the mode-multiplexer: 15 input beams from 15 single-mode fibers are reflected 15 times between phase masks and a dielectric mirror to form orthogonal modes that can be guided by the 15-mode fiber. **b** Measured wavelength-dependent loss of one mode-multiplexer. **c** Photograph of one mode-multiplexer.

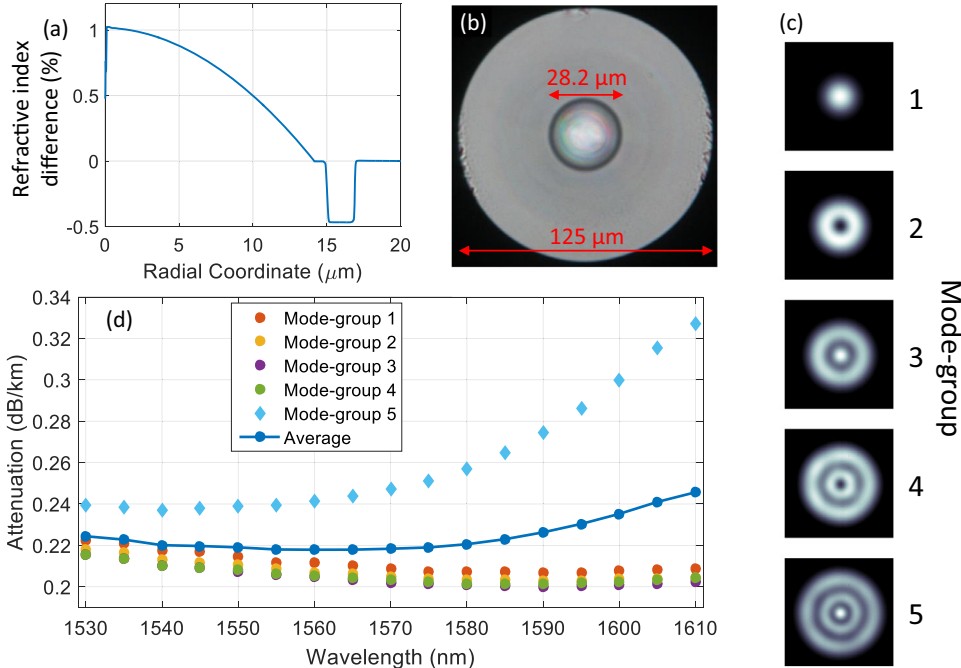

**Fig. 3 15-mode transmission fiber. a** Refractive index profile of the trench-assisted, graded-index 15-mode fiber. **b** Photograph of the fiber's cross-section. **c** Camera recording of the output mode intensity profiles after 7.6 km 15-mode fiber when excited with ASE noise of 40 nm bandwidth. Only one mode is shown per mode group, as all modes within one mode group have equal intensity patterns. Different numbers of radial intensity maxima confirm a strong mode group selectivity of mode-multiplexer and fiber. **d** Wavelength-dependent attenuation measurements of the 23 km long 15-mode fiber. The fifth mode group has a higher attenuation, attributed to increased micro-bending sensitivity.

wavelength. The fiber was fabricated with standard glass (125 μm) and coating (245 μm) diameters. Using this technique allowed to benefit from the tight process tolerances of standard MMF production[29]. The wavelength-dependent attenuation of the fiber is shown in Fig. 3d. The minimum loss, averaged over all 15 modes, was below 0.22 dB/km at 1560 nm wavelength, while the loss of the fifth mode group reached up to 0.33 dB/km at 1610 nm wavelength. Higher attenuation of the fifth mode group originates from the lower effective indexes in this mode group, increasing the micro-bending sensitivity[30]. The lowest DMGD, measured for a fiber from this manufacturing batch, was

78 ps/km at 1550 nm wavelength[31], to date the lowest reported DMGD in MMF supporting 15 modes[32,33]. A photograph of the fiber facet is shown in Fig. 3b. Figure 3c shows the measured intensity profiles after a 7.6-km span of the 15-mode MMF, when exciting one mode out of one of the five mode groups. As strong mode coupling was present between the modes in each mode group, all intensity profiles for a mode group look equal. The clear radial distinction between the five images qualitatively confirms a high mode-group selectivity of the mode-multiplexer and a strong confinement of light within a mode-group during transmission in the fiber.

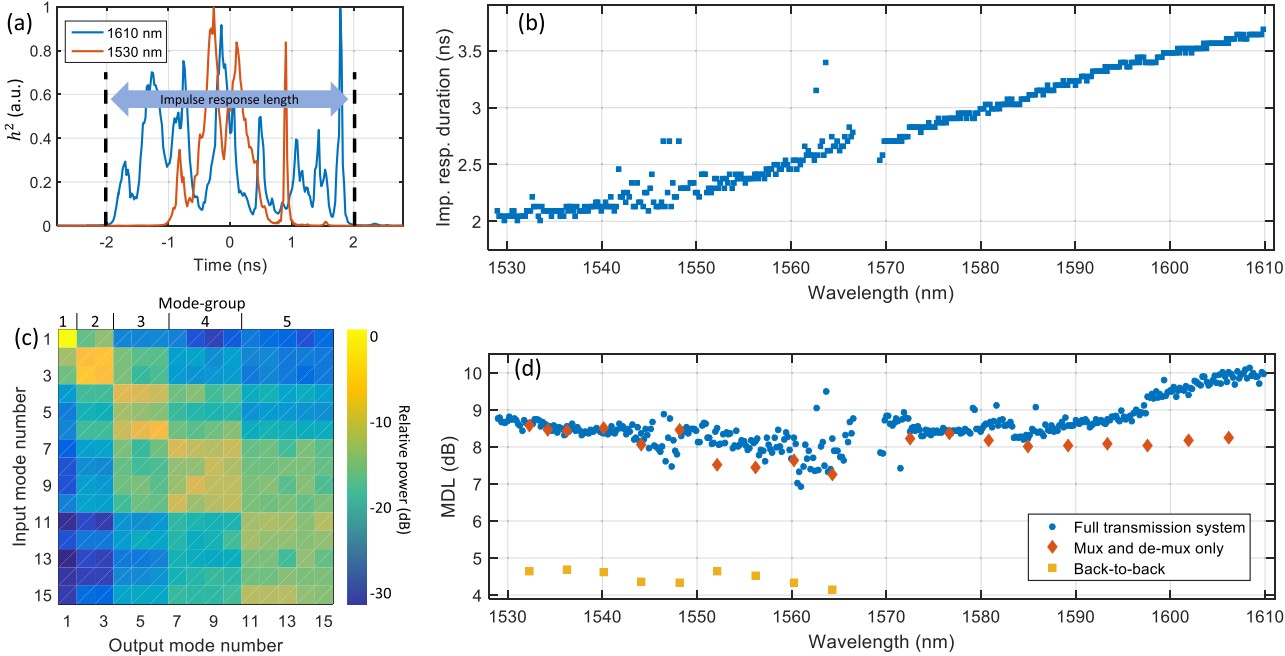

**Fig. 4 Wavelength dependence of the linear transmission properties. a** Intensity impulse responses of two wavelength channels. **b** Impulse response durations for all 382 wavelength channels. **c** Modal power coupling matrix for the wavelength channel at 1550 nm. **d** Mode-dependent loss (MDL) for all 382 wavelength channels after transmission and for selected wavelength channels in a back-to-back setup and with mode-multiplexers only.

**Wavelength-dependent linear propagation characteristics.** Figure 4a shows the intensity impulse response of two wavelength channels, one in the low C-band and one in the high L-band. The total temporal spread of the impulse response is a main factor for the complexity of the DSP[34], as the MIMO equalizer needs to cover a time window that spans the entire impulse response length. To measure the wavelength dependence of the impulse response duration, we define the impulse response length as the time interval that covers 99% of the area under the impulse response. Figure 4b shows the length of the impulse response for all 382 WDM channels. It increases from 2 ns in the low C-band up to 3.7 ns in the high L-band. This is in agreement with previous measurements and confirms the design target of the fiber[31] reaching a DMGD of less than 100 ps/km at 1550 nm wavelength, yielding an accumulated delay spread of less than 2.3 ns after 23 km transmission. Figure 4c shows the coupling matrix of the WDM channel at 1550 nm wavelength. A clear structure following the five mode-groups of the 15-mode fiber can be observed. The mode-group confinement is stronger for lower order modes, while the modes from the 4th and the 5th mode groups exhibit stronger coupling.

Different spatial paths can have different losses, resulting from characteristics such as non-perfect mode-multiplexers, connectors, or unequal fiber attenuation as shown in Fig. 3d, leading to MDL. In contrast to a long impulse response spread that can be countered by using MIMO DSP with sufficiently long memory length, MDL cannot be recovered by DSP but fundamentally lowers the capacity of the transmission system[35,36]. Figure 4d shows the MDL for selected wavelength channels in a back-to-back setup where the transmitter is directly connected to the receiver. Due to imperfections of the time-division multiplexing (TDM) receiver setup, MDL reaches 4–5 dB for C-band channels, while no data was available in the L-band. Adding the mode-multiplexers, connected by a 5 m piece of 15-mode fiber increases the MDL to 7 dB for the upper C-band and up to 9 dB in the low C- and high L-bands. Figure 4d also shows the MDL for all 382 WDM channels after 23-km transmission. When compared to

the mode-multiplexer-only values, MDL only increases toward the high L-band. This is in agreement with the fiber loss measurements in Fig. 3d, showing increased attenuation for the highest mode-group with increasing wavelength. Nevertheless, the reported MDL values confirm the broadband compatibility of the mode-multiplexers and the transmission fiber.

**Achieved data throughput.** Figure 5 shows the data rates for each of the 382 WDM spatial super-channels, calculated by using generalized mutual information (GMI)[37] and using a coding scheme[38] that was based primarily on the DVB-S2 standard[39]. The data rates of wavelength channels varied between 2.3 and 3.8 Tb/s after GMI estimation and 1.88 and 3.54 Tb/s after decoding. These changes are assumed to originate from the power variations of the comb-generated carrier-lines and enhanced phase-noise for wavelength channels at large spectral separation from the comb-seed at 1558 nm wavelength. In addition, increased loss of the highest mode-group and subsequent larger MDL values at high L-band channels contributed to a reduced channel quality. The total data rate, evaluated as the sum of the data rates for all WDM channels, was 1.01 Pb/s when using the coding scheme and 1.14 Pb/s when using GMI as performance metric.

**Discussion**
The present experiment demonstrated 1.01 peta-bit-per-second transmission in a 15-mode fiber with standard-cladding diameter. Key elements of the transmission system were an optical comb-based transmitter, covering more than 80 nm bandwidth in C- and L-bands, MPLC-based mode-selective mode-multiplexers, and a trench-assisted, graded-index 15-mode fiber. The system performance was conditioned by MDL, contributed by different parts of the transmission system. MDL was introduced by the mode-multiplexers and is assumed to stem primarily from optical misalignment. Future iterations of alignment-tolerant phase mask designs and a more industrialized assembly procedure may significantly reduce both insertion loss and MDL of the mode-

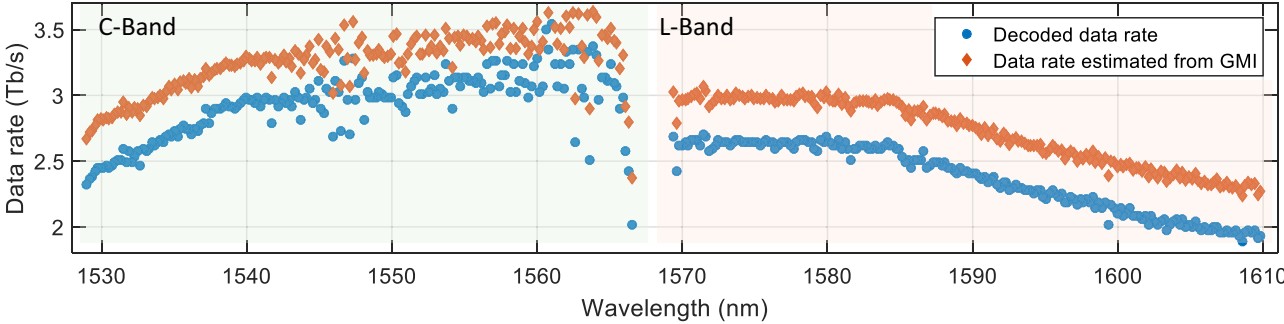

**Fig. 5 Wavelength dependence of the signal quality after transmission.** The data rate of each of the 382 spatial super-channels was estimated by an implemented coding scheme and from generalized mutual information (GMI). The C-band performed overall better compared to the L-band due to a combination of reduced mode-dependent loss and phase-noise from the comb source.

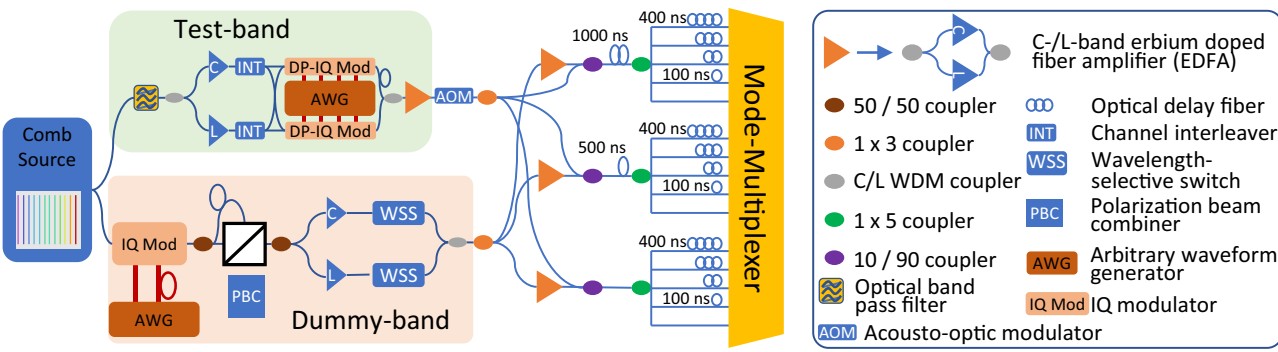

**Fig. 6 Laboratory implementation of the transmitter setup.** The output of the comb source was modulated in two sub-systems, one for a high-quality tunable test-band and one for dummy channels. Fifteen de-correlated copies of the signal were generated in a split-and-delay stage to emulate independent data channels in each fiber mode.

multiplexers. The loss of the highest order mode-group of the MMF, and consequently the MDL, increased for the high wavelength region of the L-band. While this trend is inherent to the dispersion property of graded-index MMF, additional design optimization may increase the mode field confinement of the highest order mode-group and subsequently lower MDL across the entire bandwidth. We note that an additional source of MDL arose from the TDM receiver scheme utilized to lower the required number of coherent receivers that could be eliminated by using one coherent receiver per fiber mode.

In comparison to the present work, a weakly coupled 4-core MCF transmission system has been demonstrated with a data rate of 0.61 Pb/s[22], being an alternative SDM fiber with standard cladding diameter. While that throughput was enabled by occupying an additional spectral band, it did not require higher order MIMO DSP. However, the number of spatial channels in weakly coupled MCF is limited by inter-core crosstalk to around 4–5 in a standard cladding diameter fiber. MMF, on the other hand, can be designed to guide more than 100 spatial channels in the same cladding diameter, thus enabling substantially higher data rates when fully utilizing fibers with such high mode-count. As a step toward multi-peta-bit-per-second transmission in MMF, this transmission demonstration underlines the potential of combining high channel-count wavelength and space-division multiplexing in MMF for future high-capacity optical transmission systems.

## Methods
**Experimental setup for the transmission demonstration**. The implementation of the transmitter is shown in Fig. 6. A single optical comb source generated more than 500 carrier laser lines with a 25 GHz channel spacing[40]. A total of 382 of those laser lines were used to generate 382 WDM channels covering more than 80 nm

bandwidth between 1528 and 1610 nm wavelength across the C- and L-bands. The optical spectrum was separated into a three-channel tunable test-band and a 379 channel dummy-band. A tunable optical bandpass filter selected three adjacent comb lines for the test-band. The three lines were split into even and odd channels in interleavers for modulation in two dual-polarization IQ-modulators. The modulators were driven by a 4-channel arbitrary waveform generator that was operating at 49 GSample/s, generating 24.5 Gbaud DP-64-QAM signals, based on 12 pseudo random binary sequences of length $2^{16}$. The signals were root-raised cosine shaped with a roll-off factor of 0.01. Odd and even channels were optically de-correlated by 20 ns. The remaining comb spectrum was modulated in a single-polarization IQ-modulator, combined with a polarization multiplexing emulation stage to generate a band of WDM dummy channels. C- and L-band channels were amplified in erbium-doped fiber amplifiers before wavelength-selective switches were used to equalize the power spectrum of the dummy channels and to carve a notch for accommodation of the test-band. All WDM channels were measured in turns by sequentially tuning the test-band optical filter and the notch in the dummy-band to the corresponding wavelength. In absence of 15 identical modulation sub-systems, independent data signals for transmission over the 15 fiber modes were emulated in a split-and-delay structure. By delaying the 15 split arms by multiples of 100 ns, 15 data signals were generated that were de-correlated within the memory length of the transmission system and thus can be considered as locally independent. The acousto-optic modulator (AOM) in the test-band path was required for the receiver implementation as detailed in the following section. The signals were amplified in EDFAs and launched at approximately 20 dBm per SDM channel into the mode-multiplexer.

A TDM receiver setup[41] was employed as depicted in Fig. 7a. A schematic of the TDM receiver principle is shown in Fig. 7b: three time-slots of 16.66 μs duration were allocated to one signal each. This allowed capturing the output signal streams from all 15 fiber modes with a total of five coherent receivers. The TDM receiver required time-gating of the received signals, implemented on the transmitter side with an AOM that was operated with a period of 50 μs and a duty-cycle of 33.33%. The 15 output signals from the mode de-multiplexer were sorted into five groups of three signals each. Two of the three signals in each group were delayed by 16.66 and 33.33 μs, respectively, and combined in power couplers. The five signal streams were optically amplified in EDFAs, and optical bandpass filters selected a WDM channel under test. The optical signals were received in five coherent receivers, where they were mixed with the light of a 60 kHz linewidth local oscillator (LO) laser. To ensure equal phase-noise characteristics from the LO laser in each of the

**Fig. 7 Laboratory implementation of the 15-mode receiver. a** Groups of three output ports from the mode de-multiplexer were combined in power couplers after delaying two ports by 16.66 and 33.33 μs, respectively. This enabled the reception of all 15 output ports in five coherent receiver sub-systems that contained optical bandpass filters to select a WDM channel under test, EDFAs and a common local oscillator laser. Symbol definitions as in Fig. 6. **b** Schematic description of the 3 × 1 time-division multiplexing receiver concept.

three time-slots of the TDM receiver, the LO-path had a similar split-and-delay structure as the signal paths. The 20 electrical signals, generated by the coherent receivers, were digitized in a real-time oscilloscope with 36-GHz electrical bandwidth and 80 GSample/s sampling rate. The oscilloscope was operated with a total trace length of 50 μs, corresponding to a time window of 16.66 μs for each of the 15 modes. As the transmission system had a total of 30 spatial and polarization channels (15 modes × 2 polarizations), 30 × 30 MIMO DSP was implemented with time-domain equalizers containing 281 half-symbol-duration-spaced taps, corresponding to a total time interval of 5.73 ns. Initial equalizer convergence was achieved by a fully supervised, data-aided least-mean squares (LMS) algorithm, before switching into a decision-directed LMS mode for signal performance assessment. The equalizer loop also accommodated a decision-directed phase recovery algorithm[42].

Two metrics were chosen to assess the signal quality after transmission and sub-subsequently the data rates that can be achieved in each wavelength channel: GMI[37] as well as the data rate after decoding. In both cases, code interleaving was assumed over time and modes within a wavelength channel, hence forming spatial super-channels. It is assumed that interleaving removes any residual memory seen in the channel. While GMI gives the highest data rate assuming ideal codes and bit-wise decoding, the implemented coding scheme gives a more realistic estimate of the achievable data rate. The coding scheme applied a Monte Carlo approach, where random binary patterns were generated and encoded using low-density parity check codes with different rates from the DVB-S2 standard[39] in conjunction with code rate puncturing for a code rate granularity of 0.01. A bit-to-symbol mapping was then performed before randomly selecting matching symbols from a data set constructed from the experimentally received symbol streams from all modes for the specific wavelength channel. The code rate was iteratively decreased until a post-FEC BER of less than $2.18 \cdot 10^{-5}$ minus a 10% margin was reached. A minimum of $2 \cdot 10^6$ symbols were used in the coding scheme for sufficient error statistics. An additional outer hard-decision FEC scheme was subsequently assumed with an overhead of 2.8%[43] to ensure error-free transmission performance. The post-FEC data rate was calculated by deducting the overhead of both, the inner and outer FEC from the raw data rate.

**Characterization of mode-multiplexer and transmission fiber**. A wideband ASE source, followed by a tunable optical filter and an EDFA generated 3-nm wide ASE noise, centered between 1530 and 1610 nm wavelength. This was connected to a 1 × 16 optical switch where the first 15 ports were connected to the 15 input ports of the mode-multiplexer and the remaining port was used as power reference. The output of the mode-multiplexer was a 2-m piece of 15-mode fiber with nominally equivalent characteristics to the fiber used for transmission. A free-space power meter was used to measure the output power of the mode-multiplexer for each input mode. The fiber attenuation was measured by splicing the 23-km transmission fiber to the output of the mode-multiplexer and subsequently measuring the output power of the transmission link with a free-space power meter.

**Calculation of the transmission channel matrix**. Impulse response and MDL can be estimated from the channel matrix[44]. While the MIMO DSP, used for signal reception, estimates the inverse of the channel matrix, the channel matrix was obtained by running the MIMO equalizer in the reverse direction. This was achieved by using the non-distorted, transmitted signals as the input to the MIMO equalizer and using the distorted, received signals as reference to calculate the LMS equalizer error. By using this fully supervised equalizer, we assume to produce a high-quality estimate of the time-domain channel matrix $h(t)$. The frequency-domain transmission channel $h(f)$ was calculated from the time-domain channel matrix through a Fourier transformation. The MDL of the frequency-domain

channel matrix $h(f)$ was extracted through a singular-value decomposition:

$$h(f) = U(f)\lambda(f)V(f), \qquad (1)$$

where $U(f)$, and $V(f)$ are unitary matrices and the diagonal matrix $\lambda(f)$ contains the singular values. MDL, as displayed in Fig. 4d, is then defined by averaging over the channel bandwidth as follows[44]:

$$MDL = \frac{\max(\text{avg}(\lambda(f)^2))}{\min(\text{avg}(\lambda(f)^2))}. \qquad (2)$$

The intensity averaged time-domain impulse response $h^2(t)$, as shown in Fig. 4a, was calculated as a sum of the squares of all 30 × 30 channel matrix windows.

## Data availability
The data that support the findings of this study are available from the corresponding author upon reasonable request. Source Data for Figs 1–5 are provided with the paper.

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

## Author contributions

G.R. designed and carried out the experiment and analyzed the data with support by B.J.P. and R.S.L.R.S.L., T.A.E., and G.R. prepared the DSP. N.K.F., M.M., H.C., R.R., and D.T.N. designed and assembled the mode-multiplexers. P.S. and F.A. designed, fabricated, and performed initial characterizations of the multi-mode fiber. G.R. drafted the manuscript with support by all co-authors. Y.A. and H.F. supervised the experiments.

## Competing interests

The authors declare no competing interests.
