## [Peer Review File · Nature Communications]

Reviewers' Comments:

Reviewer #1:

Remarks to the Author:

The author established a record high capacity mode-division-multiplexed (MDM) transmission exceeds 1 Pb/s utilizing standard cladding-diameter multi-mode fiber [20]. In this paper, they added new information about the characteristics of the 15-mode fiber and mode multiplexer/demultiplexer.

While the manuscript is well written and the results are well performed, however, the explanations are insufficient. Moreover, although the author compared the *data rate* of this work and a previous record [21], it's not fair comparison since the definitions of information metric are different. Therefore, this comparison is unfairly favorable to the author. I hope that the author explains sincerely and the revised paper will reach the publication level.

My comments are as follows.

1) Major: [In Page 2, line 37] **While promising...**

Although many transmission experiments utilizing standard-cladding and/or uncoupled MCF has been reported, why just one reference [18] is quoted here? For example, the ref. [*] reported standard-cladding uncoupled 4CF transmission over 3,00 km. I'm also wondering why you don't quote other reports utilizing standard-cladding multi-mode fiber.

[*] T. Matsui *et al.*, OECC2017, PDP2.

2) Minor: [In Page 3, line 54] **The theoretical loss...**

The method used to calculate the theoretical loss should be described. At least, some paper or textbook should be quoted.

3) Minor: [In Page 3, line 60] **It was thus possible...**

This sentence seems too long to read at one time. The author should explain why the DMGD kept be smaller.

4) Minor: [In Page 3, line 68] **...the lowest DMGD reported so far....**

Could you quote some previous researches about DMGD in multi-mode fibers? A reader would want to know the degree of an improvement.

5) Minor: [In Page 4, line 79] **The total temporal spread...**

Could you explain the relation between the temporal spread and DMGD described in the previous page?

6) Minor: [In Page 4, line 85] **This is an agreement with previous measurements...**

What is "previous measurements" ? .

7) Minor: [In Page 5, eq. (2)]

The average runs over the bandwidth of a signal, right? If so, it should be explained here to avoid confusing.

8) Minor: [In Page 5, line 103] **In ranges between 9 dB and....**

As in Fig4, could you also visualize the range of C band and L band in Fig. 3(d)?

9) Major: [In Page 5, line 109] **...of a symbol stream, corresponding to the...**

Could you explain why the random pickup from a symbol stream can be justified when calculate GMI? I guess that you assumed implicitly the bit-interleaved FEC coding over mode/polarization channels. If so, please explain it.

10) Major: [In Page 5, line 114] **The total data rate,**

In the ref.[21], the data rate of 402 Tb/s is defined as the capacity assuming a specific FEC scheme and its NGMI threshold. Therefore, the imperfection of the FEC is include. However, the data rate of 1.14 Pb/s in this paper is defined as GMI. As you know, GMI is the information rate when we assume the perfect binary FEC coding and bit-wise decoding. Therefore, the GMI is always larger than actual system capacity. Roughly speaking, the typical difference is 10 %. It is note that, in your previous research [20], this point was clear and it was no doubts.

Please use “decoded data rate” [20] as the metric rather than GMI, for fair comparison.

11) Minor: [In Page 6, line 128] **Odd- and even channels were...**

Could you add the value of the decorrelation between odd and even channels?

12) Minor: [In Page 7, line 157] **...equalizer with 281 half symbol-duration-spaced taps.**

To compare with Fig3.(a), could you add the corresponding time window in unit of second?. I guess that it is 5.73 ns. Moreover, could you discuss the relation between the time window and DMGD of the fiber?

13) Minor: [In Page 7, line 157] **The equalizer was initialized...**

Could you quote the reference about the algorithm for updating MIMO taps? It might be least mean square (LMS) or radius directed equalizer (RDE).

14) Major: [In Page 7, line 162] **Both, the impulse response....**

The explains is insufficient. Indeed, an error of channel estimation will introduce error of estimation of MDL [**]. You claimed that channel identification rather than equalization improves the accuracy of estimation of MDL. However, the basis is not clear. Could you explain kindly it? Does this technology have anything to do with transmission results?

[**] R. S. B. Ospina *et al.*, “DSP-based Mode-dependent Loss and Gain Estimation in Coupled SDM Transmission”, *OFC2020*, W2A.47.

Reviewer #2:

Remarks to the Author:

This paper reports state-of-the-art transmission experiment that exceeds 1 Pb/s.

The Reviewer think the paper is published as it is.

The impulse response length in Figure 3(b) recovers evert morning. The Reviewer would like to know whether the setup was adjustment every morning or not.

Reviewer #3:

Remarks to the Author:

The manuscript presents the first transmission exceeding 1 Pb/s in an optical fibre. This set a new record on data rate in any multimode fibre transmission. This resulted in approximately three times higher data rate compared to the previous work. In addition, the authors designed the system with a few-mode fibre which supported 9-LP modes while keeping the standard cladding diameter. It is a tremendous work to demonstrate the scalability of mode-division multiplexing with such a manufacturer-friendly multimode fibre. The study is definitely on a topic of relevance and interest to the readers of the journal.

The paper is well written thoroughly. However, current version of the manuscript is not ready for publication because it is inadequate as an extended version of the authors' previous work.

Concerns are below:

1. (Line 54) Related to Fig. 3(d), theoretical value of MDL is helpful to guess the MDL source.
2. (Line 55) The authors should show the loss of multiplexer before and after transporting the device. Without these values, the authors could not claim the transport caused the misalignment. At least, the readers may be curious about the best value of the loss for fabrication. If the authors are not confident in 100%, this sentence should be rephrased.
3. (Line 68) The fibre attenuation should be defined. The attenuation typically depends on spatial modes. If the attenuation of each mode is equal, it should be clearly mentioned.
4. (Lines 69 and 84) The fibre was designed for DMGD of <100 ps/km. From the design, DMGD was expected <2.3 ns. However, the minimum DMGD was observed around 2.5 ns. The authors should make sure the definitions of DMGD for both design and experiment were same. In addition, the authors should describe what caused the discrepancy between the design and experiment.

5. (Line 85) Fig. 3(b) indicates a dynamic fluctuation in the MMF transmission by unknown phenomena. This is informative; however, there is a lack of analysis. From the figure, it is not clear the characteristics in agreement with previous measurements and confirms the design target of the fibre. Theoretically estimated curve or previously measured results should be plot along the graph, helping to see a trend of the measured data and helping to estimate the gap from theory. Alternatively, the authors should describe estimation process in more detail. This would be meaningful as extension from [20].
6. (Line 86) Related to 5., the readers potentially disagree with "minimum delay spread at approximately 1550 nm wavelength" because the Impulse response seems minimised around 1530 nm. The authors should describe why this is reasonable.
7. (Line 109) Two million symbols were randomly picked from the 30 received symbol streams for estimating generalized mutual information. It would be better to mention the reason why the authors used a part of symbols and how the number of picked symbols was determined.
8. (Line 116) This paper compared to the previous work [21], in which the total capacity was calculated considering NGMI threshold. However, the submitted manuscript did not consider the threshold at all. The authors should declare the assumption and compared the records fairly.
9. (Line 126) The data frame length is not clearly mentioned in the manuscript.
10. (Line 137) The delay between 15 streams was taken as an enough length of 100 ns. Regarding 8., the delay between dual polarization was not mentioned. This might be important to make sure the decorrelation is properly considered.
12. (Line 143) Discussion of the setup specific penalty is inadequate thoroughly to evaluate the scalability/feasibility of this work. For instance, a time-multiplexed receiver setup was employed in the manuscript. However, the authors did not mention whether the performance deterioration existed or not.
13. (Section 4.3) The MDL was estimated rigorously from the MIMO equalizer coefficients. However, the paper did not discuss what caused how much MDL. The measured MDL for each component (multiplexers, fibre etc.) would be fruitful to discuss how could the work be improved and would be good for extending the authors previous work [20].
14. (Line 228) The reference style is different from others. The first names should be initials if according to other reference styles.

We would like to thank the reviewers for their generally positive assessment of our manuscript. Upon the recommendations, we performed further investigations of (1) the reason for the observation of the transmission system's change of total delay spread and (2) a further identification of (mode-dependent) loss sources, both by additional wavelength and mode resolved loss measurements but also by analyzing the channel matrix for back-to-back measurements, with and without the mode-multiplexers.

Regarding the first point, we have found that the delay spread variations were introduced by the time-domain multiplexed receiver setup (dynamic skew of 3.4 and 6.8 km delay fiber) and are not an effect that occurred in the transmission fiber. Hence, we could re-calibrate the corresponding DSP parameters and now plot solely the impact of the transmission fiber, showing a minimum in delay spread at 1530 nm that increases steadily towards higher wavelength channels.

The additional measurements of mode-dependent losses allow a better understanding of the performance reduction for channels towards the high L-band as we could identify a strong wavelength dependence of the attenuation in the highest mode-group. We also performed additional loss measurements of one mode multiplexer and recorded images of mode profiles directly after the mode multiplexer and after transmission, giving further insight into the stability of modes after propagation. An additional MDL analysis from channel matrix further allowed to show the impact of the TDM receiver setup (4dB MDL in back-to-back) as well as the MDL increase when only considering mode multiplexer and de-multiplexer, indicating that for low wavelength channels, the transmission fiber adds no MDL, while additional MDL is observed towards the high L-band channels. This is in agreement with the mode- and wavelength dependent fiber attenuation measurements.

In the following, we give detailed responses to all points raised by the reviewers. Changes in the manuscript are marked in blue.

Reviewer #1 (Remarks to the Author):

1) Major: [In Page 2, line 37] While promising...

Although many transmission experiments utilizing standard-cladding and/or uncoupled MCF has been reported, why just one reference [18] is quoted here? For example, the ref. [*] reported standard-cladding uncoupled 4CF transmission over 3,00 km. I'm also wondering why you don't quote other reports utilizing standard-cladding multi-mode fiber.

[*] T. Matsui et al., OECC2017, PDP2.

Response: We added the suggested reference to the additional demonstration using a 4-core MCF. Further demonstrations with multi-mode fibers are referenced later in the manuscript.

2) Minor: [In Page 3, line 54] The theoretical loss...

The method used to calculate the theoretical loss should be described. At least, some paper or textbook should be quoted.

Response: We added a discussion on the expected and measured loss, including mode-dependent loss of the multiplexers:

While the transformation can be loss-less when assuming ideal phase masks, fabrication limitations such as pixelated phase masks and a discretization of phase values lead to a theoretical loss of each multiplexer of approximately 0.3 dB. In addition, an excess loss of 0.25 dB can be assumed for each reflection on a phase mask due to scattering of light to higher order modes that are not supported by the transmission system, blurring of neighboring pixel values and limited reflection on the dielectric mirror, for a total loss of 3.5 dB [24].

3) Minor: [In Page 3, line 60] It was thus possible...

This sentence seems too long to read at one time. The author should explain why the DMGD kept be smaller.

Response: We changed the entire abstract describing the fiber and hope that the new version is more precise.

4) Minor: [In Page 3, line 68] ...the lowest DMGD reported so far....

Could you quote some previous researches about DMGD in multi-mode fibers? A reader would want to know the degree of an improvement.

Response: We added further references and comparisons to previously reported fibers guiding 15 modes:

The lowest DMGD, measured for a fiber from this manufacturing batch was 78 ps/km at 1550 nm wavelength [28], to date the lowest reported DMGD in MMF supporting 15 modes [29, 30].

5) Minor: [In Page 4, line 79] The total temporal spread...
Could you explain the relation between the temporal spread and DMGD described in the previous page?

Response: As this fiber operates in a weak coupling regime, the total delay spread should be approximately the product of the DMGD and the lengths of the fiber. Thus, it should be below 2.3 ns, for the wavelength with shortest DMGD. We added the following text to the manuscript for clarification:

This is in agreement with previous measurements and confirms the design target of the fiber [28] reaching a DMGD of less than 100 ps/km at 1550 nm wavelength, yielding an accumulated delay spread of less than 2.3 ns.

6) Minor: [In Page 4, line 85] This is an agreement with previous measurements...
What is "previous measurements" ? .

Response: We added a reference to a previously reported DMGD measurement of this fiber.

7) Minor: [In Page 5, eq. (2)]
The average runs over the bandwidth of a signal, right? If so, it should be explained here to avoid confusing.

Response: Yes, the averaging is performed of the signal bandwidth. We added a comment in the methods section:

MDL is then defined by averaging over the channel bandwidth as...

8) Minor: [In Page 5, line 103] In ranges between 9 dB and....
As in Fig4, could you also visualize the range of C band and L band in Fig. 3(d)?

Response: After adding more data points to Fig (now 4) (d), we added the ranges of C and L bands but found the resulting graph to be not as clear and thus prefer to leave the figure as is.

9) Major: [In Page 5, line 109] ...of a symbol stream, corresponding to the...
Could you explain why the random pickup from a symbol stream can be justified when calculate GMI? I guess that you assumed implicitly the bit-interleaved FEC coding over mode/polarization channels. If so, please explain it.

Response: Indeed, the used method implies interleaving over modes and time. We extended the signal quality assessment part in the methods section to clarify this point:

To assess the signal quality after transmission and sub-subsequently the data rates that can be achieved in each wavelength channel, we have chosen two metrics: generalized mutual information (GMI) [34] as well as the data rate after decoding as detailed in [35]. In both cases, code interleaving was assumed over time and modes within one wavelength channel, hence forming spatial-super channels. We assume that the interleaving is removing any residual memory seen in the channel. While GMI gives the highest data rate assuming ideal codes and bit-wise decoding, the implemented coding scheme gives a more realistic data rate. For the coding scheme, we apply a Monte Carlo approach, where random binary patterns are generated and encoded using low-density parity check (LDPC) codes with different rates from the DVB-S2 standard [36] in conjunction with code rate puncturing for a code rate granularity of 0.01. We then performed a bit-to-symbol mapping before randomly selecting matching symbols from a data-set constructed from the experimentally received symbol streams from all modes for the specific wavelength channel. The code rate is iteratively decreased until a post-FEC BER of less than $2.18 \cdot 10^{-5}$ minus a 10% margin is reached. A minimum of $2 \cdot 10^6$ symbols were used in the coding scheme for a sufficient error statistic. We then assume an additional outer hard-decision FEC scheme with an overhead of 2.8% [39] to ensure error-free performance. The post-FEC data rate is calculated by deducting the overhead of both the inner and outer FEC from the raw data rate.

10) Major: [In Page 5, line 114] The total data rate,
In the ref.[21], the data rate of 402 Tb/s is defined as the capacity assuming a specific FEC scheme and its NGMI threshold. Therefore, the imperfection of the FEC is include. However, the data rate of 1.14 Pb/s in this paper is defined as GMI. As you know, GMI is the information rate when we assume the perfect binary FEC coding and bit-wise decoding. Therefore, the GMI is always larger than actual system capacity. Roughly speaking, the typical difference is 10 %. It is note that, in your previous research [20], this point was clear and it was no doubts. Please use "decoded data rate" [20] as the metric rather than GMI, for fair comparison.

Response: As recommended by the reviewer, we added a second metric for the data rate assessment, as in our ECOC 2020 publication to allow a better comparison to previously published data rate records. We further added a section in the methods part where we explain the implementation of the coding scheme.

11) Minor: [In Page 6, line 128] Odd- and even channels were...
Could you add the value of the decorrelation between odd and even channels?

Response: Odd- and even channels were decorrelated by 150 ns. We added a comment to the text:

Odd- and even channels were optically de-correlated by 150ns.

12) Minor: [In Page 7, line 157] ...equalizer with 281 half symbol-duration-spaced taps. To compare with Fig3.(a), could you add the corresponding time window in unit of second?.
I guess that it is 5.73 ns. Moreover, could you discuss the relation between the time window and DMGD of the fiber?

Response: indeed, the time window of the MIMO equalizer is approximately 5.7 ns. The MIMO equalizer window needs to be at least as long as the accumulated delay spread, which is in this system the product of the DMGD and the link distance, or around 2.3 ns at 1530 nm wavelength. We added the following comment in the text:

...with half symbol-duration-spaced taps, corresponding to a total time interval of 5.73 ns.

and

This is in agreement with previous measurements and confirms the design target of the fiber [28] reaching a DMGD of less than 100 ps/km at 1550 nm wavelength, yielding an accumulated delay spread of less than 2.3 ns.

13) Minor: [In Page 7, line 157] The equalizer was initialized...
Could you quote the reference about the algorithm for updating MIMO taps? It might be least mean square (LMS) or radius directed equalizer (RDE).

Response: The algorithm for updating the data aided and the decision-directed equalizers was LMS. We added a comment in the text:

The equalizer was initialized in a supervised, data-aided mode before switching into a decision-directed mode for signal performance assessment, while both equalizers used the least-mean squares (LMS) algorithm to update the equalizer taps.

14) Major: [In Page 7, line 162] Both, the impulse response...
The explains is insufficient. Indeed, an error of channel estimation will introduce error of estimation of MDL [**]. You claimed that channel identification rather than equalization improves the accuracy of estimation of MDL. However, the basis is not clear. Could you explain kindly it? Does this technology have anything to do with transmission results?
[**] R. S. B. Ospina et al., "DSP-based Mode-dependent Loss and Gain Estimation in Coupled SDM Transmission", OFC2020, W2A.47.

Response: We used the channel matrix to calculate MDL. We do so, instead of using the equalization matrix, to stay in line with MDL definitions used by other authors. We added clarifying comments to the section and also added a reference to the extended paper by Ospina et al. (JLT, 2020). In Ref [**], it is found that MDL estimates from MIMO DSP equalizers can have an error if the SNR is low and the MDL is high. In the present experiment, we operate in a moderate SNR and moderate MDL regime where we can assume the error from MIMO-DSP based MDL estimation as acceptable. We changed the text to:

The impulse response and MDL estimations can be estimated from the channel matrix as detailed in [40]. While the MIMO DSP used for signal quality calculation estimates the inverse of the channel matrix, we calculate the channel matrix by running the MIMO equalizer in the reverse direction. This is done by using the non-distorted, transmitted signal as the input to the MIMO equalizer and using the distorted, received signal as reference to calculate the LMS equalizer error. By using this fully supervised equalizer we assume to receive a high-quality estimate of the channel matrix.

Reviewer #2 (Remarks to the Author):

This paper reports state-of-the-art transmission experiment that exceeds 1 Pb/s. The Reviewer think the paper is published as it is.

The impulse response length in Figure 3(b) recovers every morning. The Reviewer would like to know whether the setup was adjustment every morning or not.

Response: No optical adjustments were made in between the measurements. However, we also found that the day-to-day variations of the impulse response durations were an artefact from the dynamic skew of the delay

fibers in the TDM receiver setup that could be removed in the DSP, as can be seen in Figure 4(b) of the revised manuscript.

Reviewer #3 (Remarks to the Author):

1. (Line 54) Related to Fig. 3(d), theoretical value of MDL is helpful to guess the MDL source.

Response: A theoretical estimation of the MDL is not straight forward and also goes beyond the scope of this paper. However, we added further measurements of the mode-resolved losses of one multiplexer and the fiber. We also added a discussion on the theoretically expected total loss from the multiplexer:

While the transformation can be loss-less when assuming ideal phase masks, fabrication limitations such as pixelated phase masks and a discretization of phase values lead to a theoretical loss of each multiplexer of approximately 0.3 dB. In addition, an excess loss of 0.25 dB can be assumed for each reflection on a phase mask due to scattering of light to higher order modes that are not supported by the transmission system, blurring of neighboring pixel values and limited reflection on the dielectric mirror, for a total loss of 3.5 dB [24]. Figure 2(b) shows the measured wavelength-dependent average, minimum and maximum insertion loss for all 15 ports of one mode-multiplexers. The lowest average insertion loss was 9.2 dB at 1555 nm wavelength. We assume that the discrepancy between expected and measured loss stems from optical misalignment within the mode-multiplexer. Figure 2(b) also shows the loss difference between the highest and lowest loss mode, being between 2.5 dB at 1610 nm wavelength and 3.2 dB at 1530 nm wavelength. While this metric can serve as an indication of the mode-dependent loss (MDL) behavior, it should not be confused with MDL calculated from the transfer matrix, as presented later in this manuscript. A photo of one of the two used multiplexers is shown in figure 2(b).

2. (Line 55) The authors should show the loss of multiplexer before and after transporting the device. Without these values, the authors could not claim the transport caused the misalignment. At least, the readers may be curious about the best value of the loss for fabrication. If the authors are not confident in 100%, this sentence should be rephrased.

Response: As we don't have multiplexer losses measured before shipment, we have to admit that our claim was speculative. We thus removed it and instead added a wavelength dependent loss measurement for one of the multiplexers.

3. (Line 68) The fibre attenuation should be defined. The attenuation typically depends on spatial modes. If the attenuation of each mode is equal, it should be clearly mentioned.

Response: We added a mode-resolve attenuation measurement of the fiber to the manuscript. The measurement shows strongly wavelength-dependent attenuation for the highest mode-group (0.1 dB over 80 nm wavelength range vs. 0.03 dB for the 4 lowest order mode-groups). We added the following text to the manuscript:

The wavelength dependent attenuation of the fiber is shown in Figure 3(d). The minimum average loss was below 0.22 dB/km at 1560 nm wavelength, while the loss of the fifth mode group reached up to 0.33 dB/km at 1610 nm wavelength. This behavior originates in lower effective indexes of the modes in the highest mode group that increase the micro-bending sensitivity [27].

4. (Lines 69 and 84) The fibre was designed for DMGD of <100 ps/km. From the design, DMGD was expected <2.3 ns. However, the minimum DMGD was observed around 2.5 ns. The authors should make sure the definitions of DMGD for both design and experiment were same. In addition, the authors should describe what caused the discrepancy between the design and experiment.

Response: After removing the delay impact from the TDM receiver setup, we measured total delay spread of just over 2 ns at 1530 nm wavelength and approximately 2.3ns at 1550nm wavelength. This is in agreement with the design target of the fiber of DMGD of less than 100 ps/km.

5. (Line 85) Fig. 3(b) indicates a dynamic fluctuation in the MMF transmission by unknown phenomena. This is informative; however, there is a lack of analysis. From the figure, it is not clear the characteristics in agreement with previous measurements and confirms the design target of the fibre. Theoretically estimated curve or previously measured results should be plot along the graph, helping to see a trend of the measured data and helping to estimate the gap from theory. Alternatively, the authors should describe estimation process in more detail. This would be meaningful as extension from [20].

Response: We have found that the dynamic delay fluctuations were an artifact from the TDM receiver setup (dynamic skew of 3.4km and 6.8km delay fiber). Hence, it was possible to account for those dynamics in a DSP calibration. After this calibration, no day-to-day variations have been observed.

6. (Line 86) Related to 5., the readers potentially disagree with “minimum delay spread at approximately 1550 nm wavelength” because the impulse response seems minimized around 1530 nm. The authors should describe why this is reasonable.

Response: Indeed, the minimum delay spread was observed at 1530nm wavelength. We corrected this in the manuscript.

7. (Line 109) Two million symbols were randomly picked from the 30 received symbol streams for estimating generalized mutual information. It would be better to mention the reason why the authors used a part of symbols and how the number of picked symbols was determined.

Response: The method assumes an FEC interleaving over modes and time. We added further clarification on this in the methods section:

To assess the signal quality after transmission and sub-subsequently the data rates that can be achieved in each wavelength channel, we have chosen two metrics: generalized mutual information (GMI) [34] as well as the data rate after decoding as detailed in [35]. In both cases, code interleaving was assumed over time and modes within one wavelength channel, hence forming spatial-super channels. We assume that the interleaving is removing any residual memory seen in the channel. While GMI gives the highest data rate assuming ideal codes and bit-wise decoding, the implemented coding scheme gives a more realistic data rate. For the coding scheme, we apply a Monte Carlo approach, where random binary patterns are generated and encoded using low-density parity check (LDPC) codes with different rates from the DVB-S2 standard [36] in conjunction with code rate puncturing for a code rate granularity of 0.01. We then performed a bit-to-symbol mapping before randomly selecting matching symbols from a data-set constructed from the experimentally received symbol streams from all modes for the specific wavelength channel. The code rate is iteratively decreased until a post-FEC BER of less than $2.18 \cdot 10^{-5}$ minus a 10% margin is reached. A minimum of $2 \cdot 10^6$ symbols were used in the coding scheme for a sufficient error statistic. We then assume an additional outer hard-decision FEC scheme with an overhead of 2.8% [39] to ensure error-free performance. The post-FEC data rate is calculated by deducting the overhead of both the inner and outer FEC from the raw data rate.

8. (Line 116) This paper compared to the previous work [21], in which the total capacity was calculated considering NGMI threshold. However, the submitted manuscript did not consider the threshold at all. The authors should declare the assumption and compared the records fairly.

Response: We added a second metric for the data rate assessment, as in our ECOC 2020 publication to allow a better comparison to previously published data rate records. We further added a section in the methods part where we explain the implementation of the coding scheme. Please also refer to our response to comment 7.

9. (Line 126) The data frame length is not clearly mentioned in the manuscript.

Response: The trace length for each mode was 16.66 μ s for a total trace length of 50 μ s. We added this information to the manuscript:

... and a total trace length of 50 μ s, corresponding to a time window of 16.66 μ s for each mode.

10. (Line 137) The delay between 15 streams was taken as an enough length of 100 ns. Regarding 8., the delay between dual polarization was not mentioned. This might be important to make sure the decorrelation is properly considered.

Response: The test-band signals were generated with independent sequences in both polarizations with a 4-channel AWG, rather than a polarization multiplexing stage.

12. (Line 143) Discussion of the setup specific penalty is inadequate thoroughly to evaluate the scalability/feasibility of this work. For instance, a time-multiplexed receiver setup was employed in the manuscript. However, the authors did not mention whether the performance deterioration existed or not.

Response: Indeed, the used time-multiplexed receivers adds loss (e.g. 3x1 coupler, 33% duty cycle of AOM) and especially channel-dependent loss, due to the 3.4km and 6.8km delay fibers. Those impairments are not inherent to the 15-mode transmission system. To give some more insight into the various sources of impairments, we added MDL estimations to figure 4(d) for two more scenarios: back-to-back, where we directly connected the 15 decorrelated signal streams to the 15 input ports of the TDM receiver and a scenario adding mode multiplexer and de-multiplexer with 5 m of 15-mode fiber in between. While the MDL for the back-to-back case should be 0dB, we already start with more than 4dB throughout the C-band due to the TDM receiver. Unfortunately, we didn't take any traces in the L-band but think that it is reasonable to assume similar behavior in the L-band, as the used fibers and couplers were chosen with low wavelength dependence. When adding the multiplexers, MDL increases strongly with a mild wavelength dependence that is consistent with the mode-resolved loss

measurements presented in fig. 2. Finally, adding the 23km transmission fiber only mildly increases the MDL in the high L-band, as expected from the measurements shown in Fig. 3, while no additional MDL was measured in the C-band.

13. (Section 4.3) The MDL was estimated rigorously from the MIMO equalizer coefficients. However, the paper did not discuss what caused how much MDL. The measured MDL for each component (multiplexers, fibre etc.) would be fruitful to discuss how could the work be improved and would be good for extending the authors previous work [20].

Response: We added information of mode-dependent loss sources (multiplexers, fiber) throughout the article. We further have estimated the MDL for a back-to-back situation and when using only the multiplexers with 5m of 15-mode fiber in between. Please see also the response to comment 12.

14. (Line 228) The reference style is different from others. The first names should be initials if according to other reference styles.

Response: We updated the references to ensure that they all have the same style.

Reviewers' Comments:

Reviewer #1:

Remarks to the Author:

Thank you for the revisions.

The quality of the paper is gratefully improved.

In particular, the discussion of the MDL is well written.

Reviewer #2:

Remarks to the Author:

The Authors properly addressed my query and corrected figure and its descriptiopn.

Reviewer #3:

Remarks to the Author:

The authors answered all the reviewers' comments correctly. The manuscript has been revised very well and will be widely interested in the community. The manuscript is ready to be published after checking spells carefully.

As an instance, a typo was found on Line 125 in Page 6, where "DVB-2S" should be "DVB-S2."

Response to reviewers' comments:

Reviewer 1:

N/A

Reviewer 2:

N/A

Reviewer 3:

We carefully checked the paper for typos and hope that they are now reduced to a minimum.